# Study on the Potential Application of *Impatiens balsamina* L. Flowers Extract as a Natural Colouring Ingredient in a Pastry Product

**DOI:** 10.3390/ijerph18179062

**Published:** 2021-08-27

**Authors:** Eleomar de O. Pires, Eliana Pereira, Márcio Carocho, Carla Pereira, Maria Inês Dias, Ricardo C. Calhelha, Ana Ćirić, Marina Soković, Carolina C. Garcia, Isabel C. F. R. Ferreira, Cristina Caleja, Lillian Barros

**Affiliations:** 1Centro de Investigação de Montanha (CIMO), Instituto Politécnico de Bragança, Campus de Santa Apolónia, 5300-253 Bragança, Portugal; eleomar.junior@ipb.pt (E.d.O.P.J.); eliana@ipb.pt (E.P.); mcarocho@ipb.pt (M.C.); carlap@ipb.pt (C.P.); maria.ines@ipb.pt (M.I.D.); calhelha@ipb.pt (R.C.C.); iferreira@ipb.pt (I.C.F.R.F.); 2Departamento Acadêmico de Alimentos (DAALM), Câmpus Medianeira, Universidade Tecnológica Federal do Paraná (UTFPR), CEP, Medianeira 85884-000, PR, Brazil; carolinacgarcia@utfpr.edu.br; 3Institute for Biological Research “Siniša Stanković”—National Institute of Republic of Serbia, University of Belgrade, 11000 Belgrade, Serbia; rancic@ibiss.bg.ac.rs (A.Ć.); mris@ibiss.bg.ac.rs (M.S.)

**Keywords:** edible flowers, nutritional composition, phenolic compounds, bioactivities, natural colorants, food industry

## Abstract

Flowers of the genus *Impatiens* are classified as edible; however, their inclusion in the human diet is not yet a common practice. Its attractive colours have stirred great interest by the food industry. In this sense, rose (BP) and orange (BO) *I. balsamina* flowers were nutritionally studied, followed by an in-depth chemical study profile. The non-anthocyanin and anthocyanin profiles of extracts of both flower varieties were also determined by high-performance liquid chromatography coupled to a diode array and mass spectrometry detector (HPLC-DAD-ESI/MS). The results demonstrated that both varieties presented significant amounts of phenolic compounds, having identified nine non-anthocyanin compounds and 14 anthocyanin compounds. BP extract stood out in its bioactive properties (antioxidant and antimicrobial potential) and was selected for incorporation in “bombocas” filling. Its performance as a colouring ingredient was compared with the control formulations (white filling) and with E163 (anthocyanins) colorant. The incorporation of the natural ingredient did not cause changes in the chemical and nutritional composition of the product; and although the colour conferred was lighter than presented by the formulation with E163 (suggesting a more natural aspect), the higher antioxidant activity could meet the expectations of the current high-demand consumer.

## 1. Introduction

Food colouring is defined as any substance of artificial or natural origin that has the ability to confer, improve or even intensify the colouring of a food [1,2]. The popularity of the colorant class in the food industry is due to its power to win over consumers by enhancing the visual aspects, from the processing and storage stages to the purchase of the final product [3,4]. However, there has been a noticeable resistance to consumers about adopting artificial colorants in food formulations due to the relationship of these substances with some human symptoms and their possible side effects [1,5,6,7]. In this sense, the industry has been exploring cost-effective, safe and stable natural colour matrices to meet consumer needs [8,9].

Colorants from plant matrices such as leaves, stems, fruits and flowers have proven to be promising alternatives for the industrial sector due to their numerous positive health effects, and due to their increasing consumption worldwide [1,10,11]. However, the search for more stable, safe and habitable natural matrices is still a challenge to be overcome [12,13].

Due to their different attractive colours, edible flowers can be considered innovative and engaging alternatives for application in food formulations, in particular by the discovery of new compounds related to their natural pigments, such as carotenoids, chlorophylls and anthocyanins [14,15].

Many of these compounds present in flowers are known to have the ability to act against numerous human symptoms, including free radical inactivation, bactericidal and bacteriostatic activity against some strains of microorganisms and even in the remediation of inflammation and anti-aging [16].

Plants of the genus *Impatiens* are classified as edible but are mainly known for their intense cultivation in landscape interventions, and their flowers present themselves as promising matrices for the extraction of bioactive colorants due to their intense colouration [17,18].

In this sense, the present work investigated the colouring potential of the hydroethanolic extract of the pink (BP) and orange (BO) petals of the species *Impatiens balsamina* L. through the characterisation and quantification of its phenolic compounds (anthocyanin and non-anthocyanin), the study of its bioactivities, and the chemical and nutritional characterisation present in its petals. After characterisation, the extract with the greatest colorimetric and bioactive potential was applied to a Portuguese pastry product “bombocas”, in order to evaluate its behaviour as an alternative colorant compared to a control (without colouring additive) and a formulation based on strawberry gelatin (with E163 colouring additive).

## 2. Materials and Methods

### 2.1. Preparation of the Samples

The specimens of flowers of the species *I. balsamina* were collected in July and August 2019, from an urban park in Medianeira city (25°17′24.8″ S 54°05′19.2″ W) in the state of Paraná (Brazil). A portion of the plant material was identified botanically and preserved as a witness in the FLOR herbarium at the Federal University of Santa Catarina, Florianópolis, SC, Brazil. The remaining collected samples were washed in running water and deposited on absorbent paper. After drying at room temperature, the flowers were separated by colour (orange and pink). Soon after separation, they were frozen in plastic containers, lyophilised (Freezone 4.5, Labconco, Kandas City, MO, USA), crushed and finally stored in airtight bottles protected from light until further analysis.

### 2.2. Evaluation of Colour Parameters

The colouring of the fresh petals was measured using a portable colourimeter (model CR-400, Konica Minolta Sensing, Inc., Osaka, Japan), as described by Pereira et al. [19]. The values found for the CIE colour space *L** *a** *b** were recorded using illuminant C with an 8 mm diaphragm opening, which was subsequently processed according to the software “Spectra Magic Nx” (version CM-S100W 2.03. 0006), by Konica Minolta.

### 2.3. Nutritional Composition

The nutritional profile of *I. balsamina* petals was determined according to official food analysis methodologies [20]. Macronutrients were therefore quantified by analysing the content of proteins, fat and carbohydrates. Subsequently, the amounts of ash, moisture and total energy value of the samples were also determined.

### 2.4. Chemical Composition

#### 2.4.1. Sugars

The free sugar content was evaluated following the methodology previously described by Barros, Pereira and Ferreira [21]. The extraction was performed with 40 mL of ethanol 80% during 30 min at 80 °C. Then the suspension was centrifuged at 15,000× *g* for 10 min (Centurion K24OR refrigerated centrifuge, West Sussex, UK) and then concentrated under reduced pressure and defatted three times with 10 mL of ethyl ether, successively. After concentration at 40 °C, the solid residues were dissolved in water to a final volume of 5 mL and filtered through 0.2 μm Whatman nylon filters using an HPLC system coupled to a refractive index detector (Knauer, Smartline 1000 and Smartline 2300 systems, respectively). The identification of free sugars was made by high performance liquid chromatography coupled to a refraction index detector (HPLC-RI; Knauer, Smartline system 1000, Berlin, Germany) using melezitose as an internal standard. The results presented in g/100 g of fresh weight (fw).

#### 2.4.2. Fatty Acids

After extraction by Soxhlet with petroleum ether, the obtained fat extract was subjected to a methylation process with 5 mL of 2:1:1 (v/v/v) methanol/sulphuric acid/toluene in a water bath (50 °C, at 160 rpm, during 12 h); then, 3 mL of deionised water were added to obtain the phase separation. The FAME was recovered by adding 3 mL of diethyl ether, stirring on a Vortex stirrer. The sample was collected and filtered with 0.2 μm Whatman nylon filter. The fatty acids were identified by gas chromatography with flame ionisation detection (GC-FID), as previously described by Pereira, Barros, Martins and Ferreira [22]. The identification of fatty acids was made according to their relative retention times of FAME peaks of samples with known standards. CSW 1.7 software (DataApex 1.7, Prague, Czech Republic) was used to process the results, and these were expressed as a relative percentage (%) for each fatty acid detected.

#### 2.4.3. Organic Acids

Dry sample (2 g) was extracted with 25 mL of metaphosphoric acid (25 °C, 150 rpm, 45 min) and subsequently centrifuged (10,000× *g* for 5 min) and then filtered through 0.2 μm Whatman nylon filters. The organic acids were determined by high-performance liquid chromatography coupled to a photodiode detector (UFLC-PDA) according to a procedure previously described by Barros et al. [23]. The detection of organic acids was achieved using a DAD system, applying a wavelength of 215 nm (and 245 nm for ascorbic acid). The quantification of the compounds was carried out by comparing the area of their recorded peaks at the wavelengths mentioned above with the calibration curves obtained from the standards of the respective compound. The results were expressed in g/100 g (fw).

### 2.5. Phenolic Composition and Bioactive Potential of Impatiens Flower Extracts

#### 2.5.1. Extract Preparation

For the extract preparation, 0.5 g of freeze-dried petals of *I. balsamina* of both colours (pink and orange) were used. Initially, the samples were macerated at room temperature with the addition of a solution (30 mL) of ethanol/water (80:20, v/v), for 1 h (150 rpm). Subsequently, the hydroethanolic extract was filtered using a filter (Whatman No. 4), and the retained content was extracted again using the same procedure. The filtered content was deposited in a rotary evaporator (Büchi R-210, Flawil, Switzerland), and the ethanol removed under reduced pressure. Finally, the aqueous phase of both extracts was frozen and lyophilised. It should be noted that to obtain an extract rich in anthocyanins, the same methodology was used, adding 0.5% of trifluoroacetic acid (TFA) to the extraction solvent.

#### 2.5.2. Identification and Quantification of Phenolic Compounds

The identification and quantification of phenolic compounds (non-anthocyanin and anthocyanin compounds) was performed following the previously optimised methodology [24,25] and using a Dionex Ultimate 3000 UPLC system (Thermo Scientific, San Jose, CA, USA). DAD and mass spectrometer (LTQ XL mass spectrometer, Thermo Finnigan, San Jose, CA, USA) were used, working in negative mode for the detection of non-anthocyanin compounds and in positive mode for the detection of anthocyanin compounds.

Analytical curves (200–5 μg/mL) of the available phenolic standards were constructed based on the UV-Vis signal: *p*-coumaric acid (y = 30.1950*x* + 6966.7, *R*^2^ = 1, LOD = 0.68 µg/mL and LOQ = 1.61 µg/mL, peaks 1 and 2); apigenin-7-*O*-glucoside (y = 10,683*x* − 45,794, *R*^2^ = 0.996, LOD = 0.10 μg/mL; LOQ = 0.53 μg/mL, peak 3); quercetin-3-*O*-glucoside (y = 34,843*x* − 160,173, *R*^2^ = 0.9998, LOD = 0.21 µg/mL; LOQ = 0.71 µg/mL, peaks 4, 5, 6, 7, 8, and 9); pelargonidin-3-*O*-glucoside (y = 268,748*x* − 71,423; *R*^2^ = 0.9986, LOD = 0.24 µg/mL and LOQ = 0.76 µg/mL, peaks 10 to 23). The results were presented in mg/g of dry extract.

#### 2.5.3. Bioactivities Evaluation

*Evaluation of Antioxidant Activity:* To observe the antioxidant activity, the dry extract was redissolved (2.5 mg/mL) in an ethanol/water solution (80:20, v/v) and sequentially diluted to determine its EC_50_ value. The oxidative haemolysis inhibition test (OxHLIA) was performed on sheep blood samples as previously described by Lockowandt et al. [26]. In this assay, the results were expressed by their inhibitory concentrations (EC_50_ value, μg/mL) capable of producing a haemolysis delay Δ*t* of 60 and 120 min. Trolox was used as a positive control in both tests.

*Evaluation of Anti-Inflammatory Activity:* The anti-inflammatory activity was performed according to the previous methodology described by Jabeur et al. [27]. The dry extracts were dissolved in water at a concentration of 8 mg/mL and evaluated in contact with the RAW 264.7 mouse macrophage cell line. The positive control adopted was dexamethasone 50 μM and the results were expressed in terms of EC_50_ (μg/mL).

*Evaluation of the Hepatotoxic Activity:* For the cytotoxicity test, the dried extracts were diluted in water at a concentration of 8 mg/mL [23]. The cytotoxic potential was observed by using human tumour cell lines: MCF- 7 (breast adenocarcinoma), NCI-H460 (non-small cell lung cancer), HeLa (cervical carcinoma) and HepG2 (hepatocellular carcinoma) and using the sulforhodamine B assay to measure the cell growth inhibition. The hepatotoxic potential was determined using a freshly harvested porcine liver cell culture (acquired from certified slaughterhouses), designated as PLP2. Ellipticine was used as a positive control, and the results were expressed as GI_50_ values (μg/mL) as a positive control.

*Evaluation of the Antimicrobial Activity:* The dried extracts were dissolved in water (10 mg/mL), and the antibacterial potential was evaluated applying a methodology previously described by Soković, Glamoćlija, Marin, Brkić and Griensven [28]. In this assay, two Gram-negative bacteria strain *Escherichia coli* (ATCC 35210), *Pseudomonas aeruginosa* (ATCC 27853) and *Salmonella typhimurium* (ATCC 13311), and two Gram-positive bacteria strains: *Bacillus cereus* (human isolate), *Staphylococcus aureus* (ATCC 11632) and *Listeria monocytogenes* (NCTC 7973), were used. The minimum inhibitory (MIC) and minimum bactericidal (MBC) concentrations were determined using streptomycin and ampicillin as positive controls.

For the antifungal activity, the methodology described by Soković and Griensven [29] was applied and used for four fungal strains: *Aspergillus fumigatus* (ATCC 1022), *Aspergillus niger* (ATCC 6275), *Aspergillus versicolor* (ATCC 11730), *Penicillium funiculosum* (ATCC 36839), *Penicillium ochrachloron* (ATCC 9112) and *Penicillium verrucosum var. cyclopium* (food isolate). The MIC and minimum fungicidal concentration (MFC) were evaluated using ketoconazole as the positive control.

The microorganisms are deposited at Mycological Laboratory, Department of Plant Physiology, Institute for biological research “Sinisa Stanković”, University of Belgrade, Serbia.

### 2.6. Incorporation of Natural Colorant in “Bombocas”

#### 2.6.1. Formulation of the “Bombocas”

Three groups of “bombocas” samples were prepared: (i) “bombocas” Control (BC) prepared with neutral powder gelatin; (ii) Strawberry “bombocas” (BS) prepared with strawberry powder gelatin; and (iii) “bombocas” containing *Impatiens* extract (BI).

The “bombocas” were made according to a traditional recipe based on the preparation of a base syrup, obtained from the dilution of 400 g of sugar and 80 mL of glucose syrup in 120 mL of water at room temperature. After dilution, the mixture was heated to 100 °C until it reached boiling point, after which the temperature was raised to 180 °C, and the mixture remained on heating for 5 min. Simultaneously, three sheets of animal gelatine (4.2 g) and 80 g of gelatine powder were diluted in 120 mL of water for 2 min in a microwave oven (Eletric Co, MW70017SG, Guangdong, China) at the power of 1200 W. The hot diluted gelatine was beaten with an electric mixer (BRAUN, HM3135WH, Walldürn, Germany) until a thick foam developed in order to promote the incorporation of air into the mixture. The cooled base syrup was placed in contact with the gelatine foam, and 10 mL of lemon juice and the mixture was whipped at maximum speed with a power of 500 W until the marshmallow consistency was obtained. For the BI formulation, in this last step, 600 mg of *I. balsamina* extract was added. All groups of samples were analysed in triplicate, immediately after preparation and after three and seven days of storage kept at room temperature and protected from light, with each group consisting of 12 marshmallows. After freezing, all samples were lyophilised and then ground, and the mass was homogenised.

#### 2.6.2. Evaluation of Colour Parameters, Nutritional Composition, Sugar and Fatty Acid Content and Antioxidant Activity of “Bombocas” during Storage Time

The colour of the samples was assessed in triplicate, at three different points of each sample, according to the procedure described in Section 2.3.

The nutritional compositions as well as the sugar and fatty acid content of all samples from all times were determined following the procedures described previously in Section 2.3, Section 2.4.1 and Section 2.4.2, respectively.

The antioxidant activity of all “bombocas” samples, from all times, was evaluated following the methodology described previously in Section 2.5.3.

### 2.7. Statistical Analysis

All the assays were performed in triplicate, and the results expressed in the mean ± standard deviation (SD) format. An independent-samples t-test was used to classify the two extracts. The data from the “bombocas” was analysed using a two-way analysis of variance (ANOVA) followed by a Tukey’s test for homoscedastic samples and a Tahmane T2 for non-homoscedastic sample. A significance of 0.05 was used for all analysis throughout the whole manuscript (SPSS v. 23.0; IBM Corp., Armonk, NY, USA).

## 3. Results

### 3.1. Evaluation of Colour Parameters

The flowers of the genus *Impatiens balsamina* present extremely attractive colours, ranging from white, yellow and orange to more reddish tones, which in turn are directly associated with the presence of numerous compounds, especially the anthocyanins class [17]. Thus, the chromatic analysis in the CIE colour space *L** (brightness), *a** (green/red) and *b** (blue/yellow) was performed on the petals of specimens of the species *I. balsamina*, in which the results are expressed as shown in Table 1. Regarding the orange petals (BO), it was possible to verify that only the values obtained for *L** do not present statistically significant differences. In turn, the differences in the *a** and *b** parameters resulted in higher values for *I. balsamina*. Regarding the pink flowers (BP), these showed statistically significant differences for *L** and b* values justified by the slightly darker and brighter shade of BP that was observed and recorded in Table 1. Reports on the colourimetry of *Impatiens* flowers are scarce. However [30], when observing petals of the species *Impatiens walleriana* L., the orange petals presented *L** and *b** values higher than those presented by the pink variety, translating to light and yellowish, while the pink petals presented higher values in the *a** parameter, suggesting a more intense tone in the red range, a fact that corresponded to the information also obtained by this work.

### 3.2. Nutritional Composition

The nutritional composition of *Impatiens balsamina* L. petals in pink and orange varieties have been evaluated by the Official Food Analysis Methodologies (AOAC). Their respective amounts of protein, fat, carbohydrates and energy, as well as the values of ashes and sugars, were reported in Table 1. At first, it can be noted that the ash content was similar between the two colours of flowers studied. Meanwhile, orange-coloured petals (BO) had a higher amount of protein and fat, while pink petals (BP) stood out in terms of carbohydrates and had a higher energy value (21.1 kcal/100 g dw (BP) and 19.2 kcal/100 g dw (BO)). Regarding sugar content, fructose and glucose were found in both varieties, the latter being the majority sugar and the quantity being higher in the BP variety compared to BO.

Few studies report on the nutritional profile of the *Impatiens* genus. However, Fernandes, Casal, Pereira, Saraiva and Ramalhosa [31] state that the components commonly present in edible flowers, in general, do not differ much from the nutritional composition found in other plant organs. Among the few existing studies, Szewczyk et al. [32] investigated the presence of water-soluble polysaccharides of four distinct aquose extracts of species of *Impatiens* (*Impatiens glandulifera* Royle, *Impatiens parviflora* DC., *Impatiens balsamina* L. and *Impatiens noli-tangere*). From the practical aspects, it was demonstrated that the main sugars present are arabinose, ramnose, galactose, mannose, xylose and glucose, which varied their proportions according to the species. While for protein composition, quantities of 3.1 g/100 g dw [31] and 4.60 g/kg dw [33] were reported to be present in the species *I. walleriana*. In the bibliographical review by Fernandes et al. [31], the nutritional profiles of more than thirty species of edible flowers were grouped. Results were tabled for moisture (71.6 to 93.4%), total carbohydrates (10 to 90.20 g/100 g dw), proteins (2 to 52.3 g/100 g dw), fats (1.3 to 6.1 g/100 g dw), ash (2.6 to 15.9 g/100 g dw) and energy (75 to 465 kJ/100 g dw). In general, it can be noted that the values found in the nutritional composition of the petals studied in our research correspond to the values obtained for edible flowers in the literature.

### 3.3. Chemical Composition

The individual, saturated, monounsaturated and polyunsaturated fatty acids present in the BO and BP samples are detailed in Table 1. Twenty-four individual fatty acids were identified in the samples of BO and BP, in which the majority were: stearic (C18:0; 31.65 ± 0.09%), linoleic (C18:2n6; 20.8 ± 0.6%) and γ-linoleic (C18:3n6; 14.4 ± 0.1%) for BO, and linoleic (C18:2n6; 26.1 ± 0.3%), stearic (C18:0; 24.2 ± 0.2%) and γ-linoleic (C18:3n6; 21.4 ± 0.4%) for BP. However, some differences in the composition of fatty acids were observed in the varieties analysed, since saturated fatty acids were mainly in BO, whereas polyunsaturated acids stood out in BP, according to the following proportions: 44.9 ± 0.5% (BO) and 34.37 ± 0.06% (BP) for saturated fatty acids, 14.13 ± 0.08% (BO) and 12.9 ± 0.5% (BP) for monounsaturated fatty acids and 40.9 ± 0.6% (BO), and 52.7 ± 0.5% (BP) for polyunsaturated fatty acids. Szewczyk et al. [34] studied the lipophilic composition of the hexanoic extract of leaves, roots and seeds of the species *Impatiens glandulifera* Royle *and Impatiens noli-tangere* L. From the results obtained, ten different fatty acids were observed (Caprylic (C8:0); Capric (C10:0); Azelaic (C9:0); Palmitic (C16:0); Stearic (C18:0); Oleic (C18:1); Linoleic (C18:2) ω-6; α-linolenic (C18:3) ω-3; γ-linolenic (C18:3) and arachidonic (C20:4), as well as it was reported that polyunsaturated fatty acids had a higher percentage, with a predominance for α-linolenic, oleic and palmitic acids. Furthermore, the author also reinforced that leaves and seeds of the two studied species had higher amounts of saturated fatty acids than their own roots. The fatty acid profile of flowers of the species *I. balsamina* showed a particular discrepancy with the literature due to its greater diversity of individual fatty acids and its divergence in the percentages of saturated, monounsaturated and polyunsaturated fatty acids. However, these facts can be justified by the difference between the organs studied, the environmental conditions and also by the difference between the species.

The profile of organic acids found for orange (BO) and pink (BP) flowers of the species *I. balsamina* was represented by five distinct compounds (oxalic acid, quinic acid, malic acid, succinic acid, ascorbic acid), in which their proportions are described in Table 1. Among the organic acids identified, succinic acid was the majority in both samples, at 59.8 ± 0.9 g/100 g dw in the BP sample and 43.9 ± 0.9 g/100 g dw in the BO variety. However, oxalic, quinic and malic acids had a higher proportion for BO samples, while succinic and ascorbic acids were more evident in the BP sample. Still, the BP sample (134.0 ± 0.4 g/100 g dw) showed a total of organic acids higher than the BO sample (122.6 ± 2.1 g/100 g dw). Chua [35] researched small metabolites from the methanolic extract (50%, v/v) of stems of *I. balsamina* L. and identified thirteen different organic acids (hydroxybutyric acid, glyceric acid, fumaric acid, succinic acid, tartronic acid, malic acid, citramalic acid, dehydrossikimic acid, hydroxiterpenilic acid, quinic acid, gluconic acid, lauric acid, cafeic trihydroxyl acid). From these results, only three compounds corresponded to those quantified (quinic acid, malic acid, succinic acid) in our study. This justifies the need for further study on the quantification of organic acids of the genus *Impatiens*.

### 3.4. Identification and Quantification of Phenolic Compounds

Polyphenols are secondary metabolites that are present in various plant matrices, including plants, fruits, flowers, seeds, bark, leaves and roots [36]. There has been a growing interest from the investigative and industrial sector about this class of compounds due to their numerous bioactive properties presented [37]. With this perspective, the chromatographic data of each peak in terms of retention times (Tr), wavelength of maximum absorption in the UV-Vis region (max), pseudo molecular ion ([M-H]^−^/[H]^+^), and the fragmentation of the molecular ion (MSn) were used for the identification of the non-anthocyanin and anthocyanin compounds.

Regarding the non-anthocyanin group, nine compounds were tentatively identified, including two phenolic acids (*O-p*-coumaroyl) and seven flavonoids (*O*-glycosylated derivatives of apigenin, quercetin and kaempferol). Peaks 1 and 2 ([M-H]^−^ at *m/z* 325) presented a UV-Vis spectrum very characteristic of *p*-coumaric acid, with a maximum between 308 nm and 325 nm, and were tentatively identified as *O-p*-coumaroyl-α-hexoside and *O-p*-coumaroyl-β-hexoside, respectively, following the chromatographic characteristics previously described by Jaiswal and Kuhnert [38] in fruits of *Lagenaria siceraria* Stand.

Regarding the detected flavonoids, only one *O*-glycosylated apigenin was identified (peak 3, [M-H]^−^ at *m/z* 517, apigenin-*O*-malonyl-hexoside), showing MS^2^ fragments at *m/z* 311 and *m/z* 269, and its tentative identification was confirmed by comparison with the previously described in *Cynara cardunculus* var. Altilis [39]. Two *O*-glycosylated quercetin derivatives were also detected in *I. balsamina* samples, peaks 5 ([M-H]^−^ at *m/z* 667) and 6 ([M-H]^−^ at *m/z* 595) being tentatively identified as quercetin-acetyl-*O*-hexoside and quercetin-*O*-hexoside-pentoside, respectively [40]. The most representative aglycone of the flavonoid group is kaempferol, of which four *O*-glycosylated derivatives were found: peak 8 (kaempferol-3-*O*-glucoside) identification was carried out by comparing the retention time and UV-vis spectra together with the available standard; peak 4 ([M-H]^−^ at *m/z* 609), 7 ([M-H]^−^ at *m/z* 651) and 9 ([M-H]^−^ at *m/z* 543), kaempferol-*O*-hexoside-*O*-hexoside, kaempferol-acetyl-*O*-hexoside-*O*-hexoside and kaempferol-*O*-hexoside-*O*-deoxyhexoside, respectively, were previously described by by Sut et al. [41] and Ning et al. [42] in *Paeonia* species and *Cyclocarya paliurus* tea leaves, respectively.

Table 2 shows a markedly *O*-glycosylated flavonoid-rich profile and only two phenolic acids derived from *p*-coumaric acid, with kaempferol-acetyl-*O*-hexoside-*O*-hexoside (peak 7) being the majority compound in this sample (2.23 ± 0.03 and 4.014 ± 0.004 mg/g extract, in BO and BP, respectively). The pink variety (BP) showed higher content in phenolic compounds, 17.19 mg/g extract, respectively, mainly due to the content in flavonoids and phenolic acids, respectively.

Fourteen anthocyanin compounds were tentatively identified, corresponding to six glycosylated pelargonidin derivatives, six malvidin derivatives and two peonidin derivatives. Anthocyanin glycosyl acylation decreases the polarity of the acylated anthocyanins 13 to 23; consequently, their retention time in a reversed-phase column was increased. This occurs due to the change of the molecular size and spatial structure of the anthocyanin aglycone [43]. The pelargonidin derivatives were the compounds found with the highest numerical expression in the samples. The attempt to identify peak 10 ([H]^+^ at *m/z* 595), pelargonidin-*O*-dihexoside was performed accordingly as previously described by Pires et al. [44] in *Vacinnum mirtylus* L. The attempt to identify 12/15 ([H]^+^ at *m/z* 741) as pelargonidin-*O*-hexoside-*O*-deoxyhexoside-hexoside was carried out as previously described by Li et al. [45] in hydrolysate of vegetable extracts. The attempted identification of peaks 17/19 ([H]^+^ at *m/z* 783) as pelargonidin-*O-p*-coumaroyl-hexoside-*O*-acetyl-hesoxide was made following the previously described by Hosokawa et al. [46] in red flowers of *Hyacinthus orientalis*. For peak 11, also a glycosylated derivative of pelargonidin, no bibliographic reference was found with its characterisation, so identification was carried out by the presence of a pseudo molecular ion at [H]^+^ at *m/z* 637, with subsequent losses of MS^2^ fragments at *m/z* 475 (162 u) and 271 (pelargonidin aglycone, 162 u + 42 u) corresponding to the loss of one hexose residue and acetyl and hexose residues, respectively, being therefore tentatively identified as pelargonidin-*O*-hexoside-*O*-acetylhexoside. Regarding malvidin derivatives, peaks 13, 14 and 16 ([H]^+^ at *m/z* 801), tentatively identified as malvidin-3-*O-p*-coumaroylhexoside-*O*-hexoside isomer I, II and III, respectively, had already been characterised and identified by other authors [47] in red wine grape pomace. Peak 22, malvidin-*O*-coumaroylhexoside ([M]^+^ at *m/z* 639), was also already described by the same authors [47]. For peaks 20 and 23, the identification attempt was performed only by the chromatographic data obtained, in which both presented a pseudomolecular ion at [H]^+^ at *m/z* 843, with subsequent losses of MS^2^ fragments at *m/z* 639 (loss of an acetyl residue and a hexose) and *m/z* 331 (loss of a *p-*coumaric acid residue and a hexose), being therefore tentatively identified as malvidin-*O*-acetylhexoside-*O*-coumaroylhexoside. Finally, for peonidin derivatives, no bibliographic references were found to support the attempted identification of these peaks either, so the chromatographic data obtained were used. Peaks 18 and 21 were tentatively identified as peonidin-*O*-acetylhexoside-*O-p*-coumaroylhexoside, showing a pseudomolecular ion at [M]^+^ at *m/z* 813, with subsequent losses of MS^2^ fragments at *m/z* 609 (162 u + 42 u, acetyl and hexose residues) and 301 (peonidin aglycone, and 146 u + 162 u, which corresponds to *p-*coumaric acid and hexose residues).

Contrary to what was observed for the non-anthocyanin phenolic compounds, it was observed that at the level of anthocyanins, the samples presented a very similar qualitative and quantitative profile. However, similarly to what was observed for the non-anthocyanin phenolic compounds, it is again in the pink varieties of the plants under study that the highest concentration of anthocyanin compounds was obtained (18.9 ± 1.3 mg/g extract), mainly due to the presence of malvidin-3-*O*-coumaroylhexoside-*O*-hexoside derivatives, respectively (peak 7a).

### 3.5. Bioactivities Evaluation

The Oxidative Hemolysis Inhibition (OxHLIA) analysis method was used to verify the existence of antioxidant activity in the BP and BO extracts. Thus, it was possible to notice that both extracts possessed excellent antioxidant activity (Table 3), with values of 29 ± 2 µg/mL for BP and 42 ± 2 µg/mL for BO expressed in terms of EC_50_. Oldenburg, Henning and Soendergaard [48] studied *Impatiens chinensisa* plant parts (seeds, leaves, stems, roots and flowers) by means of antioxidant assays (DPPH and ABTS), using as solvent 1% of HCl, 90% of aqueous methanol. The extracts revealed the capacity of radical scavenging in the various tissues, with the flowers exhibiting the highest amount of antioxidant capacity.

The anti-inflammatory potential of the extracts (BP and BO) was proven by in vitro measurements with macrophage cells (RAW264.7) (Table 3). Thus, it was evidenced that the pink coloured extract (163.5 ± 6.8 µg/mL, BP) obtained the most promising GI_50_ value compared to the orange-coloured extract (280.8 ± 12.4 µg/mL, BO). Paun et al. [49] evaluated the anti-inflammatory activity of hydroethanolic extract (50:50, v/v) of leaves and stems of *Impatiens noli-tangere* against the inflammatory enzymes LOX, COX-1 and COX-2. The most promising IC_50_ values were, respectively, 2.46 µg/mL (LOX), 18.4 µg/mL (COX-1) and 1.9 µg/mL (COX-2) for the nanofiltration fraction (NF), which can be used for inflammatory diseases. Pires Junior et al. [30] in turn observed the anti-inflammatory activity of hydroethanolic extract (80:20, v/v) of pink and orange flowers of the species *I. walleriana*, in which diagnosed values were 312.1 ± 5.5 µg/mL for the orange extract and 349.21 ± 12.8 µg/mL for the pink extract. It can be observed that the extracts BP and BO present more promising values when compared to the similar study made with the extract of flowers of the species *I. walleriana*.

For the evaluation of the antitumour activity of BP and BO extracts, a set of human tumour cell lines (MCF-7, NCI-H460, HeLa and HepG2) was used, and the results were described according to GI50 values. The toxicity of the extracts was tested for the non-tumour cell line (PLP2) up to the maximum concentration as shown in Table 3. In general, it can be affirmed that BP and BO extracts provided an antitumor activity for all tumour cell lines. However, BP was responsible for the best bioactive concentrations, showing values of 90.4 ± 5.5 µg/mL (HeLa), 134.9 ± 9.2 µg/mL (HepG2), 154.9 ± 14.5 µg/mL (MCF7) and 167.2 ± 12.5 µg/mL (NCI-H460) when compared with the values obtained by the extract BO. Furthermore, both extracts showed no toxicity against PLP2 cell lines (GI_50_ > 400 μ g/mL). Ding et al. [50] evaluated the antitumor activity of the chloroform and ethanolic extracts of *I. balsamina* flowers and subsequently tested on HePG2 tumour cells, in which the IC_50_ values corresponded to 6.08 ± 0.08 µg/mL, while Wang et al. [51] investigated the behaviour of naphthoquinone (MeONQ) isolated from the aerial parts of *I. balsamina* against adenocarcinoma cells (MKN45), which proved that the cytotoxic activity on the cells was comparable to amoxicillin, with an IC_50_ value of 4.52 µg/mL.

The antibacterial activity of ethanolic extracts obtained from the flowers of *I. balsamina* was tested against a panel of Gram-positive (Bacillus cereus; *Staphylococcus aureus* and *Listeria monicytogenes*) and Gram-negative (Escherichia coli; Pseudomonas aeruginosa and Salmonella typhimurium) bacteria. Next, the antifungal activity of *Impatiens* extracts (BO) and (BP) was also tested for a panel of fungi (*Aspergillus fumigatus*; *Aspergillus versicolor*; *Aspergillus niger*; *Penicillium funiculosum*; Penicillium ochrachloron; *Penicillium verrucosum* var. *cyclopium*), in which the results obtained for each extract in MIC, MBC and MFC were expressed as shown in Table 3. Previous studies reported that methanolic extracts of *Impatiens* (80:20, v/v) show significant antibacterial activity against Gram-positive strains without acting effectively against Gram-negative bacteria [52]. These results are inconsistent with those obtained in this study since the Gram-negative bacteria *Escherichia coli* is overall the most sensitive for the two extracts tested (BO) and (BP). In another study, Yang et al. [53], after isolating the bioactive compound, 2-methoxy-1,4-naphthoquinone (MNQ), from the hydroethanolic extract (95%) of the aerial parts of *I. balsamina* L., tested its antifungal capacity against a panel of 8 fungi and found that all the microorganisms were susceptible. This was confirmed in our study since the extracts BP and BO presented themselves as promising for fighting fungi since the CMF values were significantly lower than CMB.

The phenolic composition referred to in Table 2, evidenced a higher number of phenolic compounds for the BP extract, which will justify the better bioactive performance of the extract in relation to BO.

### 3.6. Incorporation of Natural Colorant in “Bombocas”

The pastry is a culinary segment essentially linked to the production of sweets with attractive products. In the pastry industry, marshmallows stand out as an airy and smooth product, formulated from the combination of gelatin, sugar, glucose syrup and flavouring ingredients, highly appreciated by consumers [54,55].

The potential colouring of the extract (BP) obtained from pink flowers of *I. balsamina* (the most promising extract in terms of bioactivity) was analysed in the marshmallow-based filling for the preparation of a Portuguese pastry sweet, popularly known as “bombocas”. Thus, the “bombocas” made with the addition of *Impatiens* extract (BI) were analysed in comparison with two other formulations: a control formulation (BC) with a filling without colouring additive and another similar to the traditional recipes (BS) made with strawberry gelatine and additive E163, where the results obtained over a period of seven days are reported in Figure 1.

The influence of the incorporation of *I. balsamina* extract (BI) in the shelf life of the “bombocas” and in the centesimal profile of the marshmallow was evaluated in comparison with the other formulations (BC and BS). Regarding the nutritional value of the “bombocas” (Table 4), the most abundant nutrient are carbohydrates, followed by moisture and protein. Regarding the statistical treatment, the results were treated using a two-way ANOVA, allowing the individualised understanding of each of the factors, type of colorant (TC) and time interval (TI). After analysing the interaction between the two factors (TC × TI), if TC × TI < 0.05, they were analysed simultaneously, and general tendencies can be extracted from the Estimated Marginal Means (EMM) plots. If TC × TI > 0.05, each factor was evaluated independently and classified according to the post-hoc test. For the centesimal composition, a significant interaction was sought for all assays, meaning that both factors, namely the colorant and storage time, influenced the outcome. Still, some general tendencies were extracted from the estimated marginal means for proteins and carbohydrates (Figure 2a,b). The samples incorporated with *Impatiens* showed a higher quantity of proteins, which did not vary much over the seven days, while the control sample showed a lower value which decreased over time, revealing that the *Impatiens* extract, beyond providing proteins, also shows the potential of preserving them over storage time. In terms of the carbohydrates, the EMM plots reveal that once again, the control sample showed the least and generally reduced their quantity over the storage time, while the samples incorporated with *Impatiens* and strawberry showed a higher quantity. While the *Impatiens*-incorporated sample did reduce quantity over time, the strawberry incorporated sample maintained it. To date, there are few literature data reporting the nutritional profile of marshmallows. However, Periche, Heredia, Escriche, Andrés and Castelló [56] studied the substitution of isomaltulose in marshmallows and stated that the recommended moisture range for this class of products tends to vary between 15–22 g of water/100 g. In turn, Yudhistira, Affandi and Nusantari [57] observed the effects of adding spinach (*Amaranthus tricolor* L.) and tomato (*Solanum lycopersicum*) on the physical, chemical and sensory properties of marshmallows, reporting moisture values between 11.71% and 17.56% and ash values between 0.22% and 0.44%. Thus, in this work, the moisture of the marshmallows was higher than the values pointed out by the cited literature. However, a certain similarity regarding the ash content was confirmed.

A recent study conducted by de Oliveira Melo et al. [58] aimed to develop strawberry gelatine gummies enriched with *Hibiscus sabdariffa* L. extract (acidified aqueous solution, citric acid (1%)), verifying that the replacement of the strawberry pulp by the anthocyanin extract of Hibiscus did not influence in general the nutritional profile (protein, lipid, total fat and carbohydrate content) of the product. However, it was noted that hibiscus enrichment was able to promote changes in pH, acidity, total solids content, ash and moisture. A different phenomenon was observed in the present study, as it is possible to observe in Figure 2a,b; due to the addition of *Impatiens* flower extract, the corduroy filling showed some changes in the carbohydrate and protein, although relatively slight.

Considering the fatty acids (Table 4), palmitic acid was the most abundant fatty acid, followed by oleic, stearic and linoleic acids. As for soluble sugars (Table 4), three were detected, namely fructose, glucose and sucrose, the latter being the most abundant (Table 4). Again, a significant interaction was detected between the TC and TI, thus not being possible to classify them independently. For fructose, it was possible to draw some generic conclusions from the EMM (Figure 2c). The artificial strawberry colorant provided a higher amount of fructose for the filling of the “bombocas”. The control sample (BC) and the sample with *Impatiens* (BI) showed similar values with a similar progression over time. Again, although these small changes were observed in the fatty acid and soluble sugar profile, the changes in these profiles were residual.

The colour coordinates of the filling of the different “bombocas” throughout the seven days of storage are expressed in the *L**, *a**, *b** colour space, shown in Figure 1. Again, there was a significant interaction between the type of colorant and the time interval of the trial. Thus, some conclusions were from the EMA represented in Figure 2d. With this, it was possible to verify that the filling containing *Impatiens* extract was the most intense sample, revealing lower *L** values (lower brightness), while the control sample was the one with the highest brightness, as it was very close to the maximum value of the colour space (100). In Figure 2e, for the a* coordinates, it was possible to verify that the red colour intensity level was higher for the strawberry filling, although the filling containing Impatiens extract revealed a slightly lower red tone, losing intensity from the 3rd to 7th day. Finally, for the *b** coordinates, Figure 2f, it is possible to verify that the Impatiens extract added a greater yellow intensity to the filling, a shade that increases over the seven days of storage. The colour coordinates (*L**, *a** and *b**), when combined, form the final colour of the filling, presented in Figure 1, where the strawberry gives a bright pink colour to the “bombocas” filling, while the *Impatiens* extract shows off a softer colour which integrates well with the chocolate covering the “bombocas” and may appeal to consumers who currently tend to reject food products with solid colours by association with less healthy products. A recent study testing the enrichment of gelatine gummies with *Hibiscus sabdariffa* L. flower extract revealed that the taste and texture were not altered when compared to the original formulation, unlike the colour, which became more pronounced, proving the aqueous extract’s colouring potential [58].

The values obtained in the oxidative haemolysis inhibition assay (OxHLIA) are shown in Table 4, with the results were expressed as IC_50_ values (µg/mL) at a Δ*t* of 30 min, which translates to the concentration of the extract required to maintain 50% of the red blood cell population intact for 30 min. The BI samples showed higher antioxidant activity when compared with the BC and BS samples, considering the storage time. Moreover, IC_50_ values were higher immediately after baking, and pink balsam extract (BP) showed antioxidant activity for the marshmallow filling, although it varied with shelf life. Al-Askalany and Ghandor [59] evaluated the colouring potential of golden mulberry (*Physalis peruviana*) and beetroot (*Beta vulgaris* rubra) juice for the preparation of marshmallows. Among the results found, it was observed that the use of anthocyanin colorants resulted in an increase in antioxidant activity by 32.76% and 44.87% compared to samples with added artificial colorants and samples without colours, respectively. The cited literature reinforces the antioxidant characteristics provided by the addition of natural extracts in the composition of marshmallows, a factor that is in line with the expectations of this research since the petal extract showed antioxidant capacity when introduced into the filling of marshmallows.

Artamonova et al. [60] investigated the organoleptic, physicochemical and antioxidant properties of six different marshmallow samples according to the type of structuring and colouring agent (water, hydroethanolic extract of Sudan rose or blackcurrant). The antioxidant capacity value found in the samples with natural colorant was 2 to 2.5 times higher than the results for the samples made without the colorants. Moreover, the marshmallows made with natural colouring resisted storage for two days without any packing material and remained stable for an extended period—thirty days in airtight polyethylene packaging and carton box—with high-quality indexes and colour stability. Note that the *Impatiens* extract presented a dual functionality in marshmallow filling because, in addition to providing colour, it was also responsible for ensuring less oxidation of the filling throughout the shelf life studied (seven days), indicating that this extract can be exploited as an alternative for formulations to obtain food additives such as colorants and preservatives.

## 4. Conclusions

The colorants obtained from natural matrices have gathered great interest in the food industry; however, there are still challenges to be overcome for their exploitation and application. Many of the flowers classified as edible have attractive colours with great potential; however, further characterisation studies are still needed. In this perspective, the petals of the two varieties (pink and orange) of *I. balsamina* were chemically, bioactively and nutritionally characterised. Both varieties had bioactive potential, highlighting the pink extract that went on for testing as a colouring ingredient in the filling of a pastry product, “bombocas”. The natural ingredient gave a softer colouring to the filling when compared to the synthetic additive, providing a more natural appearance, and guaranteeing antioxidant activity throughout the shelf life without changing the chemical and nutritional composition of the food product. Thus, flowers of the *Impatiens* genus may represent auspicious colouring ingredients, with great interest in exploitation by the food industry as a new alternative to synthetic colorants.

## Figures and Tables

**Figure 1 ijerph-18-09062-f001:**
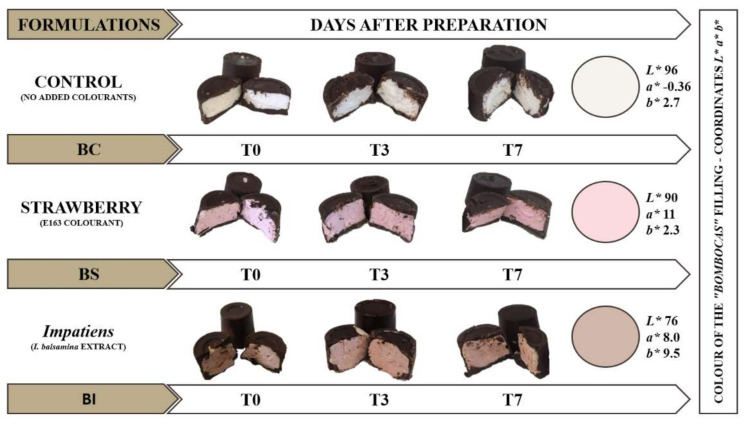
Formulations of “bombocas” made and final colouration of the filing obtained by the coordinates *L** *a** *b** (created by the author).

**Figure 2 ijerph-18-09062-f002:**
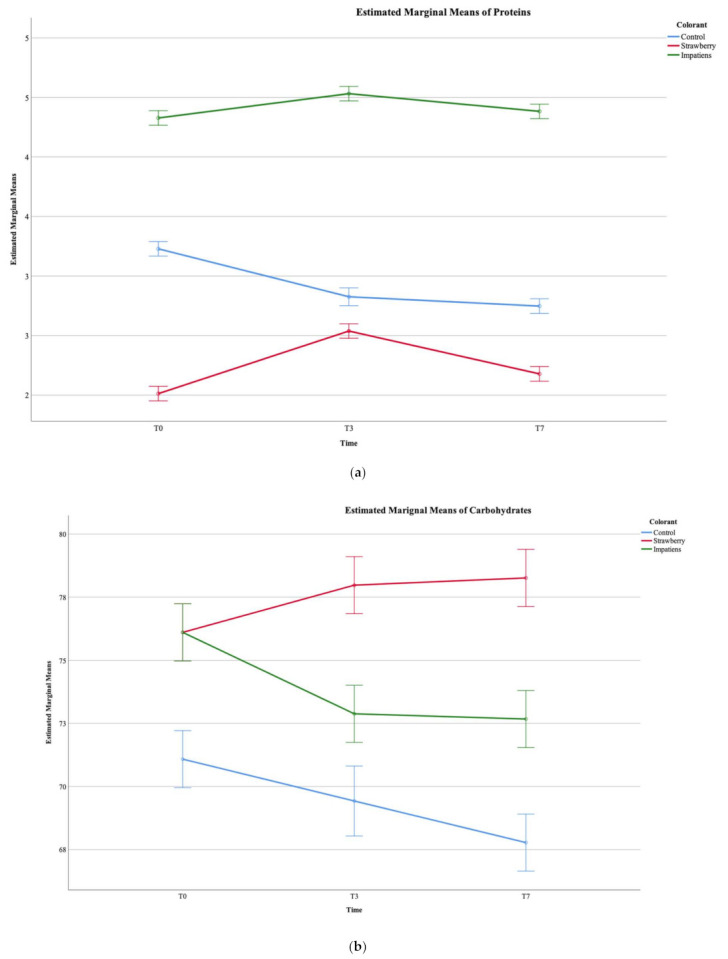
Estimated Marginal Means plots for (**a**) proteins, (**b**) carbohydrates; (**c**) fructose; (**d**) *L**; (**e**) *a**, and (**f**) *b**.

**Table 1 ijerph-18-09062-t001:** Colour parameters (CIE *L** *a** *b**), nutritional and chemical composition, fatty acids percentage (%) and quantity of organic acids (g/100 g fw) of the petals of *Impatiens balsamina* L. (orange_BO and pink_BP) (mean ± SD).

	BO	BP	*p*-Value
Colour Parameters			
*L**	46 ± 2	27 ± 1	<0.001
*a**	46 ± 2	45 ± 3	0.314
*b**	53 ± 3	14 ± 1	<0.001
Nutritional Composition			
Ash (g/100 g fw)	0.26 ± 0.02	0.26 ± 0.01	0.376
Protein (g/100 g fw)	0.33 ± 0.01	0.315 ± 0.001	0.001
Fat (g/100 g fw)	0.13 ± 0.01	0.10 ± 0.01	<0.001
Carbohydrates (g/100 g fw)	4.2 ± 0.1	4.76 ± 0.02	<0.001
Energy (kcal/100 g fw)	19.2 ± 0.4	21.145 ± 0.003	<0.001
Energy (kJ/100 g fw)	80 ± 2	88.53 ± 0.01	<0.001
Sugars			
Fructose (g/100 g fw)	0.866 ± 0.003	0.933 ± 0.001	<0.001
Glucose (g/100 g fw)	1.23 ± 0.02	1.34 ± 0.01	<0.001
Total sugars (g/100 g fw)	1.2 ± 0.02	1.34 ± 0.01	<0.001
Fatty Acids (%)			
Caprylic Acid (C8:0)	0.25 ± 0.01	0.11 ± 0.01	<0.001
Capric Acid (C10:0)	0.65 ± 0.02	0.26 ± 0.01	<0.001
Undecylic Acid (C11:0)	2.87 ± 0.04	0.90 ± 0.03	<0.001
Lauric acid (C12:0)	0.77 ± 0.03	0.38 ± 0.02	<0.001
Tridecyl acid (C13:0)	0.061 ± 0.002	0.048 ± 0.002	<0.001
Myristic acid (14:0)	2.29 ± 0.06	1.28 ± 0.06	<0.001
Myristoleic acid (C14:1)	0.97 ± 0.04	0.59 ± 0.03	<0.001
Pentadecenoic acid (C15:1)	9.8 ± 0.1	9.2 ± 0.3	<0.001
Palmitic acid (C16:0)	1.66 ± 0.07	1.04 ± 0.02	<0.001
Palmitoleic acid (C16:1)	0.51 ± 0.01	0.43 ± 0.02	<0.001
*cis*-10-Heptadecenoic acid (C17:1)	2.64 ± 0.06	2.4 ± 0.1	<0.001
Stearic acid (C18:0)	31.65 ± 0.09	24.2 ± 0.2	<0.001
Linoleic acid (C18:2n6)	20.8 ± 0.6	26.1 ± 0.3	<0.001
*γ*-linoleic acid (C18:3n6)	14.4 ± 0.1	21.4 ± 0.4	<0.001
Linolenic acid (C18:3n3)	0.56 ± 0.01	0.62 ± 0.01	<0.001
Arachidic acid (C20:0)	0.35 ± 0.02	0.37 ± 0.01	<0.001
Eicosenoic acid (20:1)	0.169 ± 0.002	0.15 ± 0.01	<0.001
Eicosadienoic acid (C20:2)	0.31 ± 0.01	0.217 ± 0.008	<0.001
Eicosatrienoic acid (C20:3n3)	3.9 ± 0.1	3.6 ± 0.2	<0.001
Dihomo-*γ*-linolenic acid (C20:3n6)	0.073 ± 0.001	0.202 ± 0.003	<0.001
Behenic acid (C22:0)	0.396 ± 0.004	2.18 ± 0.01	<0.001
Tricosanoic acid (C23:0)	3.5 ± 0.2	3.3 ± 0.1	<0.001
Tetracosanoic acid (C24:0)	0.491 ± 0.005	0.31 ± 0.01	<0.001
Tetracosenoic acid (C24:1)	0.16 ± 0.01	0.18 ± 0.01	<0.001
SFA	44.9 ± 0.5	34.37 ± 0.06	<0.001
MUFA	14.13 ± 0.08	12.9 ± 0.5	<0.001
PUFA	40.9 ± 0.6	52.7 ± 0.5	<0.001
Organic Acids (g/100 g fw)			
Oxalic acid	8.1 ± 0.7	6.13 ± 0.03	<0.001
Quinic acid	13.4 ± 0.2	11.6 ± 0.2	<0.001
Malic acid	16.7 ± 0.1	15.1 ± 0.3	0.336
Succinic acid	43.9 ± 0.9	59.8 ± 0.9	<0.001
Ascorbic acid	40.4 ± 0.3	41.3 ± 0.5	0.002
Total	123 ± 2	134.0 ± 0.4	<0.001

*L** luminosity; *a** chromatic axis from green (-) to red (+); *b** chromatic axis from blue (-) to yellow (+); fw—fresh weight, dw—dry weight; SFA—Saturated fatty acids; MUFA—Monounsaturated fatty acids; PUFA—Polyunsaturated fatty acids. Standard calibration curves: oxalic acid (y = 1E + 07*x* + 231.891, *R*^2^ = 0.9999); quinic acid (y = 671.557*x* + 14.583, *R*^2^ = 0.9998); malic acid (y = 950.041*x* + 6255.6, *R*^2^ = 0.9999); succinic acid (y = 640.365*x* − 17.602, *R*^2^ = 0.9995) and ascorbic acid (y = 4E + 07*x* + 1E + 06, *R*^2^ = 0.9909).

**Table 2 ijerph-18-09062-t002:** Retention time (Rt), wavelengths of maximum absorption in the UV-Vis region (λmax), tentative identification and quantification of phenolic compounds in *Impatiens balsamina* L. (orange_BO and pink_BP) hydroethanolic extract (mean ± SD).

**Non-Antdocyanin Phenolic Compounds**							
**Peak**	**Rt** **(min)**	**λ_max_** **(nm)**	**[M-H]^−^**	**Main Fragment** **ESI- MS^2^ [Intensity (Relative %)]**	**Tentative Identification**	**Quantification** **(mg/g)**	
						**BO**	**BP**
1	7.99	309	325	307(5),265(76), 235(100),205(5),163(5)	*O-p*-Coumaroyl-α-hexoside	0.52 ± 0.02	1.32 ± 0.05 *
2	9.13	308	325	307(5),265(82), 235(100),205(5),163(5)	O-*p*-Coumaroyl-β-hexoside	0.43 ± 0.02	1.04 ± 0.01 *
3	10.15	324	517	311(15),269(100)	Apigenin-*O*-malonyl-hexoside	1.175 ± 0.005	0.98 ± 0.02 *
4	16.07	346	609	447(21),285(100)	Kaempherol-*O*-hexoside-*O*-hexoside	1.92 ± 0.04	3.48 ± 0.02 *
5	17.13	348	667	625(100),463(10),301(20)	Quercetin-acetyl-*O*-hexoside-*O*-hexoside	0.9997 ± 0.0002	1.105 ± 0.003 *
6	17.93	342	595	301(100)	Quercetin-*O*-hexosyl-pentoside	1.27 ± 0.01	2.02 ± 0.01*
7	20.3	346	651	609(100),447(8),285(53)	Kaempherol-*O*-acetylhexoside-*O*-hexoside	2.23 ± 0.03	4.014 ± 0.004 *
8	21.94	451	447	285(110)	Kaempherol-3-*O*-glucoside	1.325 ± 0.005	1.804 ± 0.002 *
9	23.82	342	543	431(28),285(100)	Kaempherol-*O*-hexoside-*O*-deoxyhexoside	1.484 ± 0.002	1.43 ± 0.02 *
					TPA	0.95 ± 0.04	2.36 ± 0.04 *
					Tflav	10.4 ± 0.07	14.843 ± 0.005 *
					TNAC	11.4 ± 0.1	17.19 ± 0.04 *
**Antdocyanin Phenolic Compounds**							
**Peak**	**Rt** **(min)**	**λ_max_** **(nm)**	**[H]^+^**	**Main Fragment** **ESI- MS^2^ [Intensity (Relative %)]**	**Tentative Identification**	**Quantification** **(mg/g)**	
						BO	BP
10	14.62	500	595	271(100)	Pelargonidin-*O*-dihexoside	7.4 ± 0.5	1.2 ± 0.1 *
11	21.01	501	637	475(30),271(100)	Pelargonidin-*O*-hexoside-*O*-acetylhexoside	0.82 ± 0.01	0.29 ± 0.02 *
12	34.04	506	741	579(100),271(12)	Pelargonidin-*O*-hexoside-*O*-deoxyhexosyl-hexoside	0.38 ± 0.02	0.62 ± 0.03 *
13	36.62	510	801	639(25),331(100)	Malvidin-*O*-coumaroylhexoside-*O*-hexoside isomer I	0.25 ± 0.01	1 ± 0.1 *
14	37.56	511	801	639(25),331(100)	Malvidin-*O*-coumaroylhexoside-*O-*hexoside isomer II	0.26 ± 0.03	0.73 ± 0.09 *
15	38.48	504	741	579(100),271(15)	Pelargonidin-*O*-hexoside-*O*-deoxyhexosyl-hexoside	0.8 ± 0.1	0.0001 ± 0.00003 *
16	39.73	510	801	331(100)	Malvidin-*O-*coumaroylhexoside-*O*-hexoside isomer III	0.57 ± 0.02	2.7 ± 0.3 *
17	40.28	509	783	579(100),475(34),271(25)	Pelargonidin*-O*-*p*-coumaroylhexoside-*O*-acetyl-hesoxide	0.89 ± 0.09	1.3 ± 0.2 *
18	41.11	511	813	609(100),301(14)	Peonidin-*O*-acetylhexoside-*O*-*p*-coumaroylhexoside	0.43 ± 0.09^c^	0.9 ± 0.2 *
19	41.71	504	783	579(100),475(34),271(35)	Pelargonidin-*O*-*p*-coumaroylhexoside-*O*-acetyl-hexoside	2.5 ± 0.2	1.5 ± 0.1 *
20	42.7	511	843	639(100),331(34)	Malvidin-*O*-acetylhexoside-*O*-coumaroylhexoside	n.d.	2.3 ± 0.1 *
21	43.12	511	813	609(100),301(17)	Peonidin-*O*-acetylhexoside-O-coumaroylhexoside	0.7 ± 0.1	n.d. *
22	43.35	511	639	331(100)	Malvidin-*O*-coumaroylhexoside	0.8 ± 0.2	n.d. *
23	44.14	515	843	639(61),331(23)	Malvidin-*O*-acetylhexoside-*O-*coumaroylhexoside	n.d.	6.4 ± 0.5 *
					TAC	15.7 ± 0.7	19 ± 1 *

Rt—retention time; TPA—Total Phenolic Acids; Tflav—Total Flavonoids; TNAC—Total non-anthocyanin compounds; TAC—Total Anthocyanin Compounds. n.d. —not detected (below detection limit) TPC—total phenolic compounds. Standard calibration curves: *p*-coumaric acid (*y* = 301.950*x* + 6966.7, *R*^2^ = 1, LOD = 0.68 µg/mL and LOQ = 1.61 µg/mL, peaks 1 and 2); apigenin-7-*O*-glucoside (y = 10.683x − 45.794, *R*^2^ = 0.996, LOD = 0.10 μg/mL; LOQ = 0.53 µg/mL, peak 3); quercetin-3-*O*-glucoside (y = 34.843x − 160.173, *R*^2^ = 0.9998, LOD = 0.21 µg/mL; LOQ = 0.71 µg/mL, peaks 4, 5, 6, 7, 8, and 9); pelargonidin-3-*O*-glucoside (*y* = 268.748*x* − 71.423; *R*^2^ = 0.9986, LOD = 0.24 µg/mL and LOQ = 0.76 µg/mL, peaks 10 to 23). * *t*-student test *p*-value < 0.001.

**Table 3 ijerph-18-09062-t003:** Cytotoxic, hepatotoxic, anti-inflammatory, antioxidant, antibacterial (MIC and MBC mg/mL) and antifungal (MIC and MFC mg/mL) activity of the hydroethanolic extracts of *Impatiens balsamina* L. (orange_BO and pink_BP) samples (mean ± SD).

		BO		BP		Positive Control	
Antioxidant activity (Ec_50_ values; µg/mL)						Trolox	
Oxidative hemolysis inhibition assay (OxHLIA)		42 ± 2		29 ± 2 *		85.2 ± 2	
Anti-inflammatory (GI_50_ values; µg/mL)						Dexamethasone	
RAW264.7		281 ± 12		164 ± 7 *		6.30 ± 0.4	
Tumour cell lines (GI_50_ values; µg/mL)						Ellipticine	
HeLa		121 ± 3		90 ± 6 *		1.03 ± 0.09	
HepG2		201 ± 6		135 ± 9 *		1.10 ± 0.09	
MCF-7		253 ± 9		155 ± 15 *		1.02 ± 0.02	
NCI-H460		293 ± 12		167 ± 13 *		1.01 ± 0.01	
Non-tumour cell lines (GI_50_ values; µg/mL)						Ellipticine	
PLP2		>400		>400		1.40 ± 0.1	
Antibacterial activity		*B.c.*	*S.a.*	*L.m.*	*E.c.*	*P.a.*	*S.t.*
BO	MIC	0.10	0.20	0.20	0.05	0.10	0.20
	MBC	0.20	0.40	0.40	0.10	0.20	0.40
BP	MIC	0.05	0.20	0.20	0.075	0.20	0.20
	MBC	0.10	0.40	0.40	0.10	0.40	0.40
Antifungal activity		*A.fun.*	*A.v.*	*A.n.*	*P.f.*	*P.o*	*P.v.c*
BO	MIC	0.012	0.025	0.012	0.012	0.006	0.025
	MFC	0.025	0.05	0.025	0.025	0.012	0.05
BP	MIC	0.025	0.025	0.025	0.025	0.012	0.025
	MFC	0.05	0.05	0.05	0.05	0.025	0.05

GI_50_ concentration that inhibited 50% of cell growth. *B.c.: Bacillus cereus; S.a.: Staphylococcus aureos; L.m.: Listeria monocytogenes; E.c.: Escherichia coli; P.a..: Pseudomonas aeruginosa; S.t.: Salmonella Typhimirium; A.fum.: Aspergillus fumigatus; A.v.: Aspergillus versicolor; A.n.: Aspergillus niger; P.f.: Penicillium funiculosum; P.o.: Penicillium ochrochloron; P.v.c.: Penicillium verrucosum* var. *cyclopium*. * *t*-student test *p*-value < 0.001.

**Table 4 ijerph-18-09062-t004:** Nutritional, chemical, and antioxidant activity profile of the formulations Control (BC), Strawberry (BS) and *Impatiens* (BI) in relation to shelf life (0, 3, and 7 days).

			Humidity (g/100 g)		Ash (g/100 g)		Protein (g/100 g)	Fat (g/100 g)	Carbohydrates (g/100 g)		Energy (Kcal)	Energy (Kj)
Colourant Type (CT)	Control		27 ± 2		0.174 ± 0.004		2.9 ± 0.2	0.066 ± 0.004	69 ± 2		290 ± 7	1214 ± 30
	Strawberry		20 ± 1		0.191 ± 0.002		2.2 ± 0.2	0.067 ± 0.003	77 ± 2		319 ± 5	1337 ± 23
	*Impatiens*		24 ± 3		0.182 ± 0.006		4.4 ± 0.1	0.06 ± 0.04	74 ± 2		311 ± 5	1301 ± 19
*p-*value (n = 27)	Tukey Test		<0.001		<0.001		<0.001	<0.001	<0.001		<0.001	<0.001
Time Interval (TI)	T0		25 ± 2		0.181 ± 0.006		3 ± 1	0.062 ± 0.003	74 ± 2		308 ± 8	1289 ± 33
	T3		22 ± 3		0.187 ± 0.008		3 ± 1	0.069 ± 0.003	74 ± 4		310 ± 14	1296 ± 59
	T7		25 ± 4		0.180 ± 0.001		3 ± 1	0.066 ± 0.003	73 ± 5		305 ± 18	1276 ± 75
*p-*value (n = 3)	Tukey Test		<0.001		<0.001		<0.001	<0.001	0.009		0.177	0.178
TC×IT (n = 81)	*p-*value		<0.001		<0.001		<0.001	<0.001	<0.001		<0.001	<0.001
			C16:0 (%)	C18:0 (%)	C18:1 (%)	C18:2 (%)	SFA (%)	MUFA (%)	PUFA (%)	Fructose	Glucose	Sucrose
Colourant Type (CT)	Control		55 ± 4	16 ± 2	18 ± 2	10.7 ± 0.5	71 ± 2	18 ± 2	10.7 ± 0.5	12.8 ± 0.4	13.8 ± 0.3	33 ± 1
	Strawberry		49 ± 2	16.3 ± 0.7	25 ± 2	9 ± 1	65 ± 1	25 ± 2	10 ± 1	16.1 ± 0.6	16.8 ± 0.6	31 ± 1
	*Impatiens*		56 ± 5	16 ± 2	18 ± 2	10 ± 1	72 ± 3	18 ± 2	10 ± 1	13.9 ± 0.7	15 ± 1	35 ± 2
*p-*value (n = 27)	Teste Tukey		<0.001	0.008	<0.001	<0.001	<0.001	<0.001	<0.001	<0.001	<0.001	<0.001
Time Interval (TI)	T0		56 ± 4	14.6 ± 0.9	19 ± 3	10.1 ± 0.9	71 ± 3	19 ± 3	10.1 ± 0.9	14 ± 1	14 ± 1	32 ± 2
	T3		55 ± 4	15.9 ± 0.4	20 ± 4	9.2 ± 0.7	65 ± 1	20 ± 4	9.2 ± 0.7	14 ± 2	15 ± 1	33 ± 3
	T7		48 ± 2	18.2 ± 0.9	23 ± 3	10 ± 1	72 ± 3	23 ± 3	10 ± 1	15 ± 2	16 ± 1	34 ± 1
*p-*value (n = 3)	Tukey Test		<0.001	<0.001	<0.001	<0.001	<0.001	<0.001	<0.001	<0.001	<0.001	<0.001
TC × IT (n = 81)	*p-*value		<0.001	<0.001	<0.001	<0.001	<0.001	<0.001	<0.001	0.005	<0.001	0.007
OxHLIA			Control (BC)			Strawberry (BS)		*Impatiens* (BI)		Trolox		
(IC_50_, µg/mL)		T0	w.a.			124 ± 8		212 ± 29		8.8 ± 0.5		
		T3	w.a.			w.a.		267 ± 222		-		
		T7	w.a.			w.a.		486 ± 57		-		

T0—0 days; T3—3 days; T7—7 days; Palmitic acid (C16:0); Stearic acid (C18:0); Oleic acid (C18:1); Linoleic acid (C18:2); SFA—Saturated fatty acids; MUFA—Monounsaturated fatty acids; PUFA—Polyunsaturated fatty acids; w.a. —without antioxidant activity.

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
