# Peer review of "Study on the Potential Application of Impatiens balsamina L. Flowers Extract as a Natural Colouring Ingredient in a Pastry Product"

_ijerph, 2021, doi:10.3390/ijerph18179062_

Round 1

Reviewer 1 Report

Title: Study on the potential application of Impatiens balsamina L. 2 flowers as a natural colouring ingredient in a pastry product

In the present article, the authors studied the potential application of Impatiens balsamina L flowers as a natural colouring ingredient in a Portuguese pastry product, providing a chemical profil and bioactive properties about the ethanolic extract of rose (BP) and orange (BO). The non-anthocyanin and anthocyanin profile of the extracts of both flowers varieties was also determined by high-performance liquid chromatography coupled to a diode array and mass spectrometry detector (HPLC-DAD-ESI/MS). The BP extract was selected for incorporation in "bombocas" filling. Its performance as a colouring ingredient was compared with the control formulations (white filling) and with E163 (anthocyanins) colorant. The purpose of the work is confusing. It is not clear if the authors aim to isolate the dyes from Impatients flowers to be incorporated into the filling of "bombocas" or if they decided to study the hydroethanolic extract of the flowers regardless. Based on what considerations did they decide to use the ethanol-water mixture to perform the anthocyanin extraction? Why they did not use water which is more selective towards these compounds? It should be pointed out the reason why they decided to study hydroethanolic extract by inserting the appropriate references. I recommend the acceptance of the paper after major revisions listed below.

Abstract

Please, provide more details to the abstract. I remember that abstract should be able to stand alone and should summarize all results. Values, numeric results of the described tests should be added to the abstract to present the collected results; please show the type of solvent used and analysed in the study.

line 18, page 1: please write “Impatients” in italics.

line 27, page 1: “BR” should be replaced with BP.

Materials and methods

The section "materials and methods" lacks a detailed description because, even if the bibliographical references are indicated, the solvent and the extraction method of each nutrient should be immediate since in the discussion section a comparative study is made with the extracts of similar researches.

Others:

line 73, page 2: please indicate the specimen number and the collection period of flowers.

line 85, page 2: “as described by used by”  should be replaced with “as described by Pereira et al.”.

lines 105-110, page 3: Results should be calculated correctly with the calibration curves for each standard (external standard method) and expressed in mg or µg/mL rather than in  percentage. Indicate how many replicates you performed for each fatty acid analysis since you gave the result ± standard  deviation  in the results section (see page 9 lines 295-301).

line 131, page 4: the sentence “ the same methodology was adopted” is not clear.

line 161, page 4: “were deposited” should be replaced with “were dissolved”.

line 163, page 4: get rid of “in which” .

Results

line 208, page 5: insert  after “the thick foam” “in order to promote the incorporation of air into the mixture ”  and the references 54 and 55.

line 294, page 9: get rid of “with” before “stearic”

line 297, page 9: get rid of “a)” before “for BP”

line 324, page 9: what did you mean for “the methanolic extract (50:50, v/v)” ? did you mean the aqueous methanolic extract (50%, v/v)?

lines 324- 330, page 9: I think it is not possible to compare the content of two extracts obtained with different solvents. You used methanol as extracting solvente ( see reference 23), whereas  Chua( see reference 35) used water: methanol = 50:50.

Table 2: Please, write all the results with two decimal places

lines 375- 376, page 13: the sentence “Being linked to low polarity organic acids, peaks 13 to 23 showed decreasing polarity” is not correct. It should be replaced with “Anthocyanin glycosyl acylation decreases the polarity of the acylated anthocyanins 13 to 23” (see reference 43)

line 392, page 13: : “13/14/16” should be replaced with “13, 14 and 16”.

line 396, page 13: : “20/23” should be replaced with “20 and 23”.

line 402, page 13: : “18/21” should be replaced with “18 and 21”.

lines 487-491, page 14: : the sentence“The pastry is a culinary segment essentially linked to the production of sweets with 487 attractive products. In the pastry industry, marshmallows stand out as an airy and smooth 488 product, formulated from the combination of gelatin, sugar, glucose syrup and flavouring 489 ingredients, and a beating process that promotes the incorporation of air into the mixture. 490 [54, 55].”  is a repetition.  It has already been written in paragraph 2.6.1 of the materials and methods section .

Check  the words to be written in italics in the manuscript

Author Response

Reviewer #1: In the present article, the authors studied the potential application of Impatiens balsamina L flowers as a natural colouring ingredient in a Portuguese pastry product, providing a chemical profil and bioactive properties about the ethanolic extract of rose (BP) and orange (BO). The non-anthocyanin and anthocyanin profile of the extracts of both flowers varieties was also determined by high-performance liquid chromatography coupled to a diode array and mass spectrometry detector (HPLC-DAD-ESI/MS). The BP extract was selected for incorporation in "bombocas" filling. Its performance as a colouring ingredient was compared with the control formulations (white filling) and with E163 (anthocyanins) colorant. The purpose of the work is confusing. It is not clear if the authors aim to isolate the dyes from Impatients flowers to be incorporated into the filling of "bombocas" or if they decided to study the hydroethanolic extract of the flowers regardless. Based on what considerations did they decide to use the ethanol-water mixture to perform the anthocyanin extraction? Why they did not use water which is more selective towards these compounds? It should be pointed out the reason why they decided to study hydroethanolic extract by inserting the appropriate references. I recommend the acceptance of the paper after major revisions listed below.

Answer: We appreciate your careful reading and all your comments in order to improve the manuscript. The present work initially aimed to characterize two varieties of flowers belonging to the species Impatiens balsamina aiming at their potential valorization and use. After confirming the extract with the greatest bioactive potential, its incorporation into "bombocas" filling was tested. The selected solvent was chosen based on previous studies of anthocyanin extraction and with the objective of being able to use the same extract in an incorporation in the food industry. The objective was rewritten so that it was clearer.

Abstract

Please, provide more details to the abstract. I remember that abstract should be able to stand alone and should summarize all results. Values, numeric results of the described tests should be added to the abstract to present the collected results; please show the type of solvent used and analysed in the study.

Answer: We appreciate your comment. Some data were added to the abstract in order to make it more complete.

line 18, page 1: please write “Impatients” in italics.

Answer: Thanks for the comment.  The suggested change has been made.

line 27, page 1: “BR” should be replaced with BP.

Answer: Thanks for the comment. The suggested change has been made.

Materials and methods

The section "materials and methods" lacks a detailed description because, even if the bibliographical references are indicated, the solvent and the extraction method of each nutrient should be immediate since in the discussion section a comparative study is made with the extracts of similar researches.

Answer: We appreciate your comment. The methodology is presented in a reduced form and referring to official analysis methodologies or bibliographic references of articles with detailed methodology. This procedure has been adopted in order to avoid accusations of self-plagiarism that we recurrently receive when we present a more detailed methodology. In the discussion about different extracts or solvents, we were always careful to refer in detail to the difference that could justify the difference in data with those presented in the literature.

line 73, page 2: please indicate the specimen number and the collection period of flowers.

Answer: Thanks for the comment. The requested information has been added.

line 85, page 2: “as described by used by” should be replaced with “as described by Pereira et al.”.

Answer: Thanks for the comment.  The suggested change has been made.

lines 105-110, page 3: Results should be calculated correctly with the calibration curves for each standard (external standard method) and expressed in mg or µg/mL rather than in percentage. Indicate how many replicates you performed for each fatty acid analysis since you gave the result ± standard  deviation  in the results section (see page 9 lines 295-301).

Answer: All analyzes were performed in triplicate as mentioned in the methodologies described in the referenced bibliography. For fatty acids, the identification was made by chromatographic comparison of the retention times of the sample fatty acid methyl ester (FAME) peaks with those of the commercial standard 47885-U (37 FAME standard), and the results were expressed as relative percentage of each fatty acid. These compounds were expressed in relative percentage, this is the normal form to present this type of results, the identification/quantification was performed using a 37 FAME standard and individual calibration curves cannot be performed with this mixture, therefore, these compounds are usually quantified in terms of relative percentage. Moreover, in order to compare these results with literature it is easier when using this type of quantification.

line 131, page 4: the sentence “the same methodology was adopted” is not clear.

Answer: The sentence was rewritten to be clearer.

line 161, page 4: “were deposited” should be replaced with “were dissolved”.

Answer: Thanks for the comment.  The suggested change has been made.

line 163, page 4: get rid of “in which”.

Answer: Thanks for the comment.  The suggested change has been made.

Results

line 208, page 5: insert after “the thick foam” “in order to promote the incorporation of air into the mixture” and the references 54 and 55.

Answer: Thanks for the comment.  The suggested change has been made.

line 294, page 9: get rid of “with” before “stearic”

Answer: Thanks for the comment.  The suggested change has been made.

line 297, page 9: get rid of “a)” before “for BP”

Answer: Thanks for the comment.  The suggested change has been made.

line 324, page 9: what did you mean for “the methanolic extract (50:50, v/v)” ? did you mean the aqueous methanolic extract (50%, v/v)?

Answer: Thanks for the comment.  The suggested change has been made.

lines 324- 330, page 9: I think it is not possible to compare the content of two extracts obtained with different solvents. You used methanol as extracting solvente (see reference 23), whereas  Chua ( see reference 35) used water: methanol = 50:50.

Answer: In the present study, as referenced in the methodology, an ethanolic mixture was used as a solvent. In the discussion, whenever the solvent presented is different, this is referred to in order to verify whether this change can cause any difference in the results

Table 2: Please, write all the results with two decimal places

Answer: We understand the Reviewer point of view, but we believe that significant figures are important in terms of accuracy or precision. In all tables, the decimal places were set by the magnitude of the standard deviation, in order to reduce an increase of errors and oversimplification. Thus, imagining the magnitude of one standard deviation is in the grams, what would be the point of adding the average to the milligram if the standard deviation (error) is in the grams? Thus, by presenting the decimal places according to the magnitude of the standard deviation the results are clearly understood, and the quality of the work (low standard deviation) demonstrated.

lines 375- 376, page 13: the sentence “Being linked to low polarity organic acids, peaks 13 to 23 showed decreasing polarity” is not correct. It should be replaced with “Anthocyanin glycosyl acylation decreases the polarity of the acylated anthocyanins 13 to 23” (see reference 43)

Answer: Thanks for the comment.  The suggested change has been made.

line 392, page 13: “13/14/16” should be replaced with “13, 14 and 16”.

Answer: Thanks for the comment.  The suggested change has been made.

line 396, page 13: “20/23” should be replaced with “20 and 23”.

Answer: Thanks for the comment.  The suggested change has been made.

line 402, page 13: “18/21” should be replaced with “18 and 21”.

Answer: Thanks for the comment.  The suggested change has been made.

lines 487-491, page 14:  the sentence “The pastry is a culinary segment essentially linked to the production of sweets with 487 attractive products. In the pastry industry, marshmallows stand out as an airy and smooth 488 product, formulated from the combination of gelatin, sugar, glucose syrup and flavouring 489 ingredients, and a beating process that promotes the incorporation of air into the mixture. 490 [54, 55].”  is a repetition.  It has already been written in paragraph 2.6.1 of the materials and methods section.

Answer: Thanks for the comment. The sentence was changed to be clearer and less repetitive

Check the words to be written in italics in the manuscript

Answer: The entire manuscript has been checked and the changed words have been marked in yellow.

Reviewer 2 Report

This paper is a study on the potential of Impatiens balsamina L. flowers as a natural colorant, and I admire the vast amount of data. However, the amount of colorant used in food is minute, and the various components revealed in this study are not a source of nutrition for us. For the same reason, it does not exert any bio-regulatory function in the amounts used for coloring. As it is, the use of Impatiens balsamina L. flowers as a colorant misleads people into thinking that these nutrients or bioactivities are available or active. In order to overcome this point, the nutritional components and bioactivities contained in the appropriate amount used as a coloring agent should be presented. Although the above points are the most important, there are other points that should be corrected as follows, and this paper cannot be accepted as it is.
1) The references that the petals of Impatiens balsamina L. are safe for human consumption are needed.
2) In 2.6.1, the weight of one bombocas is not given, so we don't know how much coloring is in one bombocas.
3) The definitions of orange-coloured petals (BO) and while pink petals (BP) are written on line 268, but need to be where they first appear (probably in the paragraph on line 235). 
4) Since the compounds in Table 2 are identified mainly by MS (not NMR), it is difficult to determine the position of the modified carbon. However, for peaks 8 and 16, the modified positions are listed. For example, since kaempferol has four OH groups, it is necessary to deny that it is not a glycoside in other positions, citing previous literature.
5) In Table 3, there is no positive control for all items, so it is not clear whether the bioactivities are high or not. It is also necessary to consider which of the substances identified in Table 1 or 2 may be responsible for these bioactivities.
6) Before discussing the components and bioactivities in cooked bombocas, it is necessary to provide data that the dried powder of the flowers is not affected by heating or storing.

Author Response

Reviewer #2: This paper is a study on the potential of Impatiens balsamina L. flowers as a natural colorant, and I admire the vast amount of data. However, the amount of colorant used in food is minute, and the various components revealed in this study are not a source of nutrition for us. For the same reason, it does not exert any bio-regulatory function in the amounts used for coloring. As it is, the use of Impatiens balsamina L. flowers as a colorant misleads people into thinking that these nutrients or bioactivities are available or active. In order to overcome this point, the nutritional components and bioactivities contained in the appropriate amount used as a coloring agent should be presented. Although the above points are the most important, there are other points that should be corrected as follows, and this paper cannot be accepted as it is.

Answer: We appreciate the careful reading of the manuscript and your comments. The main objective of the incorporation study was to add the extract necessary to obtain a desired colour. After incorporation, it was important to prove that this incorporation does not affect the nutritional composition or sugar and fatty acid content of the product. After checking the bioactivities of the extract, it became interesting to assess whether the extract, in addition to acting as a dye, could preserve its antioxidant properties over its shelf life. This is a preliminary state that in the future will be interesting and of relevant importance for the food industry. All changes suggested below were made and marked in yellow in order to improve the manuscript.

1) The references that the petals of Impatiens balsamina L. are safe for human consumption are needed.

Answer: Thanks for your suggestion. The requested information has been added.

2) In 2.6.1, the weight of one bombocas is not given, so we don't know how much coloring is in one bombocas.

Answer: We appreciate your comment. The original recipe was divided evenly into 3 shelf time groups, totaling 12 marsmalows for each time group. Results are based on the mass of each group and not each individual drum. The information on the amount of marsmalows originated for each group was added to the manuscript in order to make the information clearer and more complete.

3) The definitions of orange-coloured petals (BO) and while pink petals (BP) are written on line 268, but need to be where they first appear (probably in the paragraph on line 235).

Answer: Thank you for your comment. The color definition has been added in the correct place

4) Since the compounds in Table 2 are identified mainly by MS (not NMR), it is difficult to determine the position of the modified carbon. However, for peaks 8 and 16, the modified positions are listed. For example, since kaempferol has four OH groups, it is necessary to deny that it is not a glycoside in other positions, citing previous literature.

Answer: I understand the reviewers concern, thus compound 8 was identified in comparison with the commercial standard, therefore, the exact position of the sugar on the aglycone is possible. Regarding compound 16, it was a misunderstanding and the sugar position on the aglycone was removed since it was identified as an isomer of compounds 13 and 14 and both these compounds also don’t have the exact position of the sugars.

5) In Table 3, there is no positive control for all items, so it is not clear whether the bioactivities are high or not. It is also necessary to consider which of the substances identified in Table 1 or 2 may be responsible for these bioactivities.

Answer: Thank you for your comment. The valuation of bioactivities was always compared with values presented by other authors in similar articles presented in the literature, which by comparison allowed us to affirm the potential of the extract. There is not just one unique compound responsible for the bioactivities, thus the most responsible compounds for the bioactivities are the phenolic compounds, since they were the compounds that were identified in the extract in which the bioactivities were evaluated. This information has been added to the results discussion in order to make it more enlightening and complete.

6) Before discussing the components and bioactivities in cooked bombocas, it is necessary to provide data that the dried powder of the flowers is not affected by heating or storing.

Answer: Thank you for your comment. The color and bioactivities of the extract were evaluated before and after addition to the mixture of ingredients and, additionally, over the shelf life in order to monitor the evolution and stability of the colorant potential. The results presented confirm the potential over shelf life in Figure 1.

Round 2

Reviewer 1 Report

I have reviewed the manuscript entitled " Study on the potential application of Impatiens balsamina L. flowers as a natural colouring ingredient in a pastry product ". The author reply properly, but minor revision are needed .

#1: Comment

My previous comment: The section "materials and methods" lacks a detailed description because, even if the bibliographical references are indicated, the solvent and the extraction method of each nutrient should be immediate since in the discussion section a comparative study is made with the extracts of similar researches.

The author’s response: We appreciate your comment. The methodology is presented in a reduced form and referring to official analysis methodologies or bibliographic references of articles with detailed methodology. This procedure has been adopted in order to avoid accusations of self-plagiarism that we recurrently receive when we present a more detailed methodology. In the discussion about different extracts or solvents, we were always careful to refer in detail to the difference that could justify the difference in data with those presented in the literature.

I think it is possible to indicate the solvent of extraction avoiding accusations of self-plagiarism and resulting at the same time more immediate for the reader the comprehension of the discussions in which you indicate the extraction solvent of other studies, but in reporting your data for the comparison you do not highlight your solvent that you might wrongly identify with the hydroalcoholic extract. In this regard I invite you to check your answer on the extraction solvent of organic acids).

#2: Comment

My previous comment: The purpose of the work is confusing. It is not clear if the authors aim to isolate the dyes from Impatients flowers to be incorporated into the filling of "bombocas" or if they decided to study the hydroethanolic extract of the flowers regardless. Based on what considerations did they decide to use the ethanol-water mixture to perform the anthocyanin extraction? Why they did not use water which is more selective towards these compounds? It should be pointed out the reason why they decided to study hydroethanolic extract by inserting the appropriate references.

The author’s response: We appreciate your careful reading and all your comments in order to improve the manuscript. The present work initially aimed to characterize two varieties of flowers belonging to the species Impatiens balsamina aiming at their potential valorization and use. After confirming the extract with the greatest bioactive potential, its incorporation into "bombocas" filling was tested. The selected solvent was chosen based on previous studies of anthocyanin extraction and with the objective of being able to use the same extract in an incorporation in the food industry. The objective was rewritten so that it was clearer.

In the new version you forgot to highlight in the abstract that it is the BP extract to be incorporated and not the BP as such. Even in the title, the term flower rather than extract seems ambiguous to me. You should properly edit it otherwise it incorrectly seems that flowers were incorporated not the extract, as it actually is

#3: Comment

My previous comment: lines 324- 330, page 9: I think it is not possible to compare the content of two extracts obtained with different solvents. You used methanol as extracting solvente (see reference 23), whereas  Chua ( see reference 35) used water: methanol = 50:50.

The author’s response: In the present study, as referenced in the methodology, an ethanolic mixture was used as a solvent. In the discussion, whenever the solvent presented is different, this is referred to in order to verify whether this change can cause any difference in the results

Your answer is not correct. In reference [23] (MATERIAL AND METHOD SECTION) samples (~2 g) were extracted by stirring with 25 mL of metaphosphoric acid (25ºC at 150 rpm) and consequently you did not compare your ethanol-water extract as you said in your reply with the methanol-water extract reported by Chua. I believe that the comparison is not correct and and so you can't assert that this justifies the need for  further study on the quantification of organic acids of the genus Impatiens.. Everywhere, please emphasise in the discussion your extraction solvent and make the appropriate considerations.

Author Response

I have reviewed the manuscript entitled " Study on the potential application of Impatiens balsamina L. flowers as a natural colouring ingredient in a pastry product ". The author reply properly, but minor revision are needed.

Answer: We appreciate your careful reading of the manuscript as well as all your comments. We are pleased that the corrections made in the previous version met your expectations. We will make the minor revisions requested and mark it in yellow in the new version.

1) My previous comment: The section "materials and methods" lacks a detailed description because, even if the bibliographical references are indicated, the solvent and the extraction method of each nutrient should be immediate since in the discussion section a comparative study is made with the extracts of similar researches.

The author’s response: We appreciate your comment. The methodology is presented in a reduced form and referring to official analysis methodologies or bibliographic references of articles with detailed methodology. This procedure has been adopted in order to avoid accusations of self-plagiarism that we recurrently receive when we present a more detailed methodology. In the discussion about different extracts or solvents, we were always careful to refer in detail to the difference that could justify the difference in data with those presented in the literature.

I think it is possible to indicate the solvent of extraction avoiding accusations of self-plagiarismand resulting at the same time more immediate for the reader the comprehension of the discussions in which you indicate the extraction solvent of other studies, but in reporting your data for the comparison you do not highlight your solvent that you might wrongly identify with the hydroalcoholic extract. In this regard I invite you to check your answer on the extraction solvent of organic acids).

Answer: We appreciate your comment. The requested information has been added to the methodology.

2) My previous comment: The purpose of the work is confusing. It is not clear if the authors aim to isolate the dyes from Impatients flowers to be incorporated into the filling of "bombocas" or if they decided to study the hydroethanolic extract of the flowers regardless. Based on what considerations did they decide to use the ethanol-water mixture to perform the anthocyanin extraction? Why they did not use water which is more selective towards these compounds? It should be pointed out the reason why they decided to study hydroethanolic extract by inserting the appropriate references.

The author’s response: We appreciate your careful reading and all your comments in order to improve the manuscript. The present work initially aimed to characterize two varieties of flowers belonging to the species Impatiens balsamina aiming at their potential valorization and use. After confirming the extract with the greatest bioactive potential, its incorporation into "bombocas" filling was tested. The selected solvent was chosen based on previous studies of anthocyanin extraction and with the objective of being able to use the same extract in an incorporation in the food industry. The objective was rewritten so that it was clearer.

In the new version you forgot to highlight in the abstract that it is the BP extract to be incorporated and not the BP as such. Even in the title, the term flower rather than extract seems ambiguous to me. You should properly edit it otherwise it incorrectly seems that flowers were incorporated not the extract, as it actually is.

Answer: We appreciate your comment. Missing information has been added to both the title and the abstract in order to convey the correct information to the reader.

3) My previous comment: lines 324- 330, page 9: I think it is not possible to compare the content of two extracts obtained with different solvents. You used methanol as extracting solvente (see reference 23), whereas Chua ( see reference 35) used water: methanol = 50:50.

The author’s response: In the present study, as referenced in the methodology, an ethanolic mixture was used as a solvent. In the discussion, whenever the solvent presented is different, this is referred to in order to verify whether this change can cause any difference in the results

Your answer is not correct. In reference [23] (MATERIAL AND METHOD SECTION) samples (~2 g) were extracted by stirring with 25 mL of metaphosphoric acid (25ºC at 150 rpm) and consequently you did not compare your ethanol-water extract as you said in your reply with the methanol-water extract reported by Chua. I believe that the comparison is not correct and and so you can't assert that this justifies the need for  further study on the quantification of organic acids of the genus Impatiens.. Everywhere, please emphasise in the discussion your extraction solvent and make the appropriate considerations.

Answer: We appreciate your comment. The manuscript was revised and the requested information regarding the solvents used by the works referenced in the discussion was added.

Reviewer 2 Report

1) By the author's answer, I understood the main objective of the incorporation study. This point should be mentioned in the text. 

2) Regarding the bioactivities in Table 3, the activity of the BP and BO extracts that you show can only be of value if it is compared to a well known substance that serves as a positive control. For example, the reference you cite for The Oxidative Hemolysis Inhibition (OxHLIA) analysis [26] shows a positive control.

Author Response

By the author's answer, I understood the main objective of the incorporation study. This point should be mentioned in the text.  

Answer: Thank you for your comment. At the end of the introduction, the sentence describing the objective was changed in order to make it clearer.

Regarding the bioactivities in Table 3, the activity of the BP and BO extracts that you show can only be of value if it is compared to a well known substance that serves as a positive control. For example, the reference you cite for The Oxidative Hemolysis Inhibition (OxHLIA) analysis [26] shows a positive control.

Answer: We appreciate your comment. The requested information regarding the values of the positive controls were referred to in the table footnote. However, in order to make the information clearer and more understandable to the reader, a column was added in Table 3 with the requested information.